# Analysis of big data job requirements based on K-means text clustering in China

Dai Debao[1], Ma Yinxia [1]*, Zhao Min[2]

1 Faculty of management, Information management and information systems, Shanghai University, Shanghai, China, 2 SHU-UTS SILC Business School, Shanghai University, Shanghai, China

* yinxia.ma@foxmail.com

## Abstract

This paper aims to understand the characteristics of domestic big data jobs requirements through k-means text clustering, help enterprises, and employees to identify big data talents, and promote the further development of big data-related research. Firstly, the crawler software is used to crawl the recruitment information about "big data" on the zhaopin.com recruitment website. Then, Jieba word segmentation and K-means text clustering are used to cluster big data recruitment positions, and the number of clustering was determined by the average sum of squares within the group. Finally, big data jobs are divided into 10 categories, and the urban distribution, salary level, education requirements, and experience requirements of big data jobs are discussed and analyzed from the perspectives of the overall data set and clustering results, to clarify the characteristics of big data job demands. The analysis results show that the job demands of big data are mainly distributed in first-tier cities and new first-tier cities. Enterprises are more inclined to job seekers with a college degree or bachelor's degree and more than one year's relevant experience. There are wage differences among different types of jobs. The higher the position, the higher the requirement for education and experience will be.

## 1. Introduction

Modern society has entered the era of big data, and data has been generally recognized as the most competitive and valuable asset at present and has become one of the important strategic resources of all countries in the world [1]. Countries around the world have formulated big data strategies to seize the commanding heights of big data development strategies. China attaches great importance to the status and role of big data in promoting economic and social development. In 2014, big data was written into the government work report for the first time. Big data has gradually become a hot topic for governments at all levels. The concepts of open sharing of government data, data circulation and trading, using big data to guarantee and improve people's livelihood have been deeply rooted in people's minds. Since then, relevant departments of the state have issued a series of policies to encourage the development of the big data industry. With the great attention and policy support of the state, the big data industry has rapidly emerged and developed, and a large number of big data-related jobs have emerged

**Data Availability Statement:** All relevant data are within the manuscript and its Supporting Information files.

**Funding:** The authors received no specific funding for this work.

**Competing interests:** The authors have declared that no competing interests exist.

at the right moment. As a result, the demand for data talents of enterprises has greatly increased. According to the 2016–2020 China Big Data Industry Panorama Survey and Development Trend Forecast Report, in 2013, the demand scale of China's big data market was 1.12 billion yuan, which was more than twice that of the previous year in 2014 and 2015, and exceeded 10 billion yuan in 2016. It is estimated that the demand scale of China's big data market is expected to reach 107 billion yuan in 2020. It can be inferred that the size of the big data market will grow exponentially in the next few years. However, the development and rise of big data technology in China have taken a short time, and the cultivation of big data talents has not kept pace with the needs of enterprises. The lack of skills and experience of big data practitioners has led to the widening gap of big data positions, and the phenomenon that big data positions are not in line with the needs has become one of the key obstacles restricting the development of China's big data industry [2]. According to the Big Data Talent Report released by Datalink Search, there are currently only 460,000 big data talents in China. The gap of big data talents in the next 3–5 years will reach 1.5 million. Among them, data analysis talents are in great demand, but the data supply index of analytical talents is low, only 0.05, which is highly scarce [3]. According to statistics from the Data Analysis Professional Committee of the China Business Federation, China's basic data talent gap will reach 14 million in the future, and more than 60% of China's Internet industry recruitment positions are recruiting big data talents [4,5]. So what data positions does the company currently have, and what specific requirements do these positions have for talents? Through the collection and statistical analysis of my country's big data talent network recruitment information, this paper sorts out the talent requirements, salary, geographical distribution, and job categories of big data-related positions, and then analyzes the demand for big data work in my country. This article analyzes the needs of big data people in depth, provides references for data talent training and job-seekers' knowledge and ability construction, and helps companies quickly locate talents. It is important for scientifically and effectively to promote big data talent training, clarify big data talent positioning, and promote big data talents. The development of the data industry has important guiding significance.

## 2. Related studies

With the booming development of the big data industry, the demand for big data jobs has been surging and gradually diversified. However, at present, there are few types of research on big data jobs, most of which use traditional statistical analysis and information measurement methods. Puncheva-Michelotti [6]. studied whether the transmission of CSR awareness in online recruitment information can improve the attractiveness of job seekers through manual statistical analysis of CSR awareness. Zhang and Zheng [7]. used descriptive statistics to analyze gender differences in self-evaluation and salary expectation in Online recruitment information in China. Papoutsoglou [8]. used multivariate statistical analysis to explore the skills and abilities of IT talents from recruitment advertisements, and analyzed the correlation between skills and abilities. Uhm [9]. and Karakatsanis [10]. used O*NET analysis software to analyze the organizational relationship, industry trend, regional trend, and other industry information among recruitment posts, providing market change information for job seekers and recruiters.

To sum up, although traditional statistics and metrology methods have made some achievements in studying the job demands of big data, the above researches have the problem of the small amount of data. At the same time, a large amount of manpower is also needed to analyze the data in the research process, and the research results are easily affected by people's cognitive level, which makes it impossible to fully excavate the laws hidden beneath the data surface.

Compared with traditional statistics and information metrology, the unsupervised learning clustering algorithm is not limited to human subjective factors and sampling data to mine job recruitment information of big data. It can mine and discover massive data and better express the potential content of data [11]. In existing studies, some scholars use the KNN algorithm [12–14] and unsupervised machine learning [15,16] to classify online recruitment information according to recruitment posts or job descriptions, to excavate labor market information or talent demand in professional fields. Besides, text mining technology [17–19] is also widely used to deeply dig the talent demand information in the job description text, to obtain the skills, technologies, or knowledge fields that the position needs to master. Among them, the semanteme-based subject model [20–23] is often used to mine job skills and requirements, extract the subject words of the job description, and then conduct manual induction, analysis, and summary of the subject words to further dig the potential demand information.

Text clustering is one of the common methods in text mining technology, which is widely used in many fields such as topic discovery and hot spot tracking [24]. In the specific implementation process, a variety of algorithms can be applied to text clustering. K-means is one of the classical clustering algorithms in the partitioning method. Due to its high efficiency, it is widely used to cluster large-scale data [25]. At present, there have been a lot of researches based on k-means text clustering. Kanungo [26] and Wu [27] improved the k-means algorithm, enhanced its superiority and operation speed, and made this method more effective in the field of text mining. Many studies have applied the k-means algorithm to document clustering [28–30]. By extracting text features, building an overlapping document clustering model, setting term weight, and other methods, the accuracy of document clustering is improved and the error between clustering is reduced. The semantically based K-means algorithm can combine the TF-IDF algorithm [31], LDA model [32], and BTM model [33] to effectively detect and mining hot topics in social networks, make up for the sparse problem of the traditional topic model, and better understand short texts. It can also classify video users in combination with emotional analysis, so as to understand the emotional similarities and differences of certain types of video viewers [34]. It can be seen that the semantic-based K-means clustering can effectively understand the passage, process natural language from all aspects, and dig out its potential information and value.

To sum up, the innovation of this paper is to obtain recruitment information related to big data on Zhaopin.com recruitment website through web crawlers. Based on the traditional K-means clustering algorithm, combining stutter segmentation and regular expression to improve the word effect, the understanding of the model to the short text is improved, and the number of clustering is determined by calculating the sum of squares within each group so that the text clustering effect is better. This model is applied to big data job recruitment information mining, replacing traditional statistics and metrology methods to obtain clustering results, and analyze the results, so as to comprehensively mine the demand characteristics of big data-related jobs in the industry.

## 3. Methods

### 3.1 Data collection

Zhaopin.com recruitment website is one of the large recruitment websites in China, which publishes a large number of recruitment information related to big data. Using "big data" as the keyword, this paper uses web crawler software to capture the recruitment information of big data-related positions in all walks of life from five dimensions, such as job title, experience requirement, education requirement, working place, and salary range. The collected data fields and examples are shown in Table 1. The data were collected from 25 cities in total, as shown in

**Table 1. Shows the collected data.**

| Data dimension | Job title | Salary range | Working place | Degree required | Experience requirements |
|---|---|---|---|---|---|
| **The sample** | Big data architect | 20,000 to 30,000 | Beijing | junior college | 5–10 years |

Table 2. Among them, first-tier cities are Beijing, Shanghai, Guangzhou, and Shenzhen, 12 new first-tier cities are Chengdu, Hefei, and Hangzhou, and 9 other cities besides first-tier and confidence cities are Dalian, Fuzhou, and Harbin. The job postings were made in three months between August 2020 and October 2020, with 16,551 online job postings.

### 3.2 Data procession

To improve the quality of unstructured text data, it is necessary to preprocess the data [35]. In this paper, first of all, the collected data were reprocessed, and 14,496 valid data were obtained after deleting invalid and incomplete data. Then, stammer segmentation was used to divide the job name data into words [36]. Since the data set contains technical terms, this paper added a self-defined dictionary in the pre-processing stage, which contains related terms for big data technology, such as SPARK, SQL, HADOOP, data analysis, data mining, and other professional terms. To ensure the accuracy of word segmentation, regular expression rules are added to match strings. Finally, the stop words list is used to automatically filter out auxiliary words, mood words, and other meaningless words. The flow chart of data preprocessing is shown in Fig 1.

### 3.3 Data analysis

K-means text clustering is widely used in many fields of text analysis. This algorithm has high efficiency and can be used to cluster large-scale data. In this paper, k-means text clustering is used to conduct text clustering analysis of the processed job name data. By combining natural language processing and clustering algorithm, the similarity within the cluster is improved and the similarity between clusters is reduced. Big data positions are divided and different big data positions are identified. In the clustering process, the sum of squares within the group after each clustering was calculated to determine the optimal number of clustering clusters. Finally, the text selected 10 clusters, and the sum of squares within the group under this number of clustering was 0.6. The similarity in the groups divided by big data positions was relatively high.

## 4. Results

There are a large number of big data positions and titles. After cluster analysis, big data positions are divided into 10 categories, and the top 5 positions in the categories are counted by word frequency. However, the number of samples in category 6 is large, so 10 high-frequency

**Table 2. Name and number of cities in each category.**

| City category | City name | City number |
|---|---|---|
| **First-tier cities** | Beijing, Shanghai, Guangzhou, Shenzhen | 4 |
| **New first-tier cities** | Chengdu, Hefei, Hangzhou, Nanjing, Shenyang, Tianjin, Wuxi, Wuhan, Xi'an, Changsha, Zhengzhou, Chongqing | 12 |
| **Others** | Dalian, Fuzhou, Harbin, Guiyang, Jinan, Ningbo, Xiamen, Shijiazhuang, Changchun | 9 |

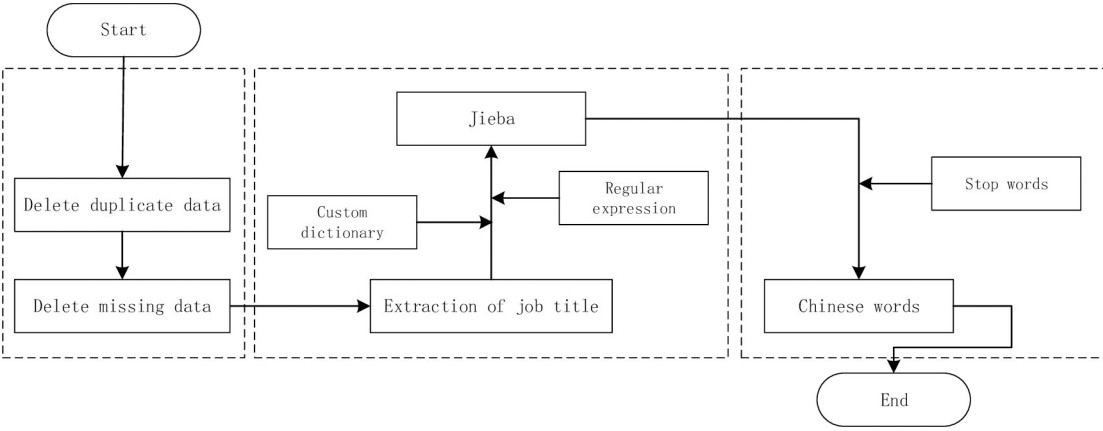

**Fig 1. Data preprocessing flow chart.**

positions are selected, as shown in Table 3. There are a large number of big data positions and titles. After cluster analysis, big data positions are divided into 10 categories, and the top 5 positions in the categories are counted by word frequency. However, the number of samples in category 6 is large, so 10 high-frequency positions are selected, as shown in Table 3. As can be seen from Table 3, the correlation and consistency between clustering keywords and high-frequency posts are relatively high. Category 1 mainly development engineer position, category 2 engineers, mainly engaged in data category 3 is the data operating post, category 4 is mainly engaged in basic data entry and management post, category 5 is greater demand for data analysis commissioner, category 6 is the biggest demand of data statistics, data annotation, data mining, such as jobs, category 7 represents the data analyst positions, including assistant, elementary, intermediate and advanced level of other position, category of 8 high frequency is a data analyst and data analysis engineer, class 9 is mainly related to data sales positions, rank higher, such as sales manager, Category 10 is the product manager position related to big data.

## 5. Analysis

To further explore the market demand for big data positions, this paper conducts a multi-dimensional and in-depth analysis from the aspects of job demand, urban distribution, educational background requirements, experience requirements, salary range, and so on.

### 5.1 The characteristics of big data jobs from the whole data set

On the whole, the demand for big data jobs is mainly distributed in first-tier cities and new first-tier cities. Among first-tier cities, Guangzhou and Shanghai have the highest demand for big data talents; among the new first-tier cities, Nanjing, Hangzhou, Xi 'an and Chengdu have the highest demand for big data jobs; while among other cities, Jinan has the highest demand for big data talents, reaching 6%, as shown in Table 4. Among 14,496 big data positions, 52.39% require a bachelor's degree and 35.13% require a junior college degree. It can be seen that a bachelor's degree is the demand subject for big data positions, followed by a junior college degree, which indicates that big data positions have low requirements for an academic degree, and most enterprises have no particularly high requirements for the academic degree of big data positions. Since the population base of master's degree and doctor's degree in China is much smaller than that of bachelor's degree and a junior college degree, the highly educated talents in the field of big data only account for 3.76%. In terms of work experience,

**Table 3. K-means clustering results and related positions.**

| Clustering categories | Sample size | Clustering keywords | Relevant position |
|---|---|---|---|
| **Cluster 1** | 2034 | development, big data, engineer, data, java, advanced, database, R&D, data warehouse, platform | Big data development engineer database<br>Development engineer<br>Big data engineer<br>Big data research and development engineer<br>Data development engineer |
| **Cluster 2** | 994 | engineer, data, data analysis, data warehouse, development, big data, data mining, database, data processing, expert | Data engineer<br>Big data operation and maintenance engineer<br>Data governance engineer<br>Data communication engineer<br>Etl engineer |
| **Cluster 3** | 534 | data operations, specialist, assistant, supervisor, manager, analysis, direction, senior, intern, engineer | Data operator<br>Data operation specialist<br>Data operation assistant<br>Data operation manager<br>Data operation supervisor |
| **Cluster 4** | 756 | data, development, engineer, direction, product, platform, data warehouse, supervisor, database, operation | Data entry clerk<br>Data clerk<br>Data development<br>Data administrator<br>Data researcher |
| **Cluster 5** | 1465 | specialist, data, data analysis, commodities, statistics, data processing, operations, assistants, sales, data labeling | Data analysis specialist<br>Data specialist<br>Commodity specialist<br>Data statistics specialist<br>Data processing specialist |
| **Cluster 6** | 5122 | data, data processing, engineers, assistants, clerks, sales, statisticians, data annotations, architects, interns | Data statistician<br>Data clerk<br>Data annotator<br>Project manager<br>Statistician<br>Data supervisor<br>Data warehouse engineer<br>Big data architect<br>Data processing engineer<br>Data mining engineer<br>Gis data processing engineer |
| **Cluster 7** | 1248 | analyst, data, finance, big data, senior, assistant, direction, operations, intern, market | Data analyst<br>Big data analyst<br>Financial data analyst<br>Senior data analyst<br>Data analyst assistant |
| **Cluster 8** | 1352 | data analysis, engineer, assistant, sales, intern, clerk, finance, direction, operation, supervisor | Data analyst<br>Data analyst<br>Data analysis engineer<br>Data analysis assistant<br>Sales data analyst |
| **Cluster 9** | 674 | manager, sales, data, data analysis, big data, advanced, data center, data management, marketing, operations | Sales manager<br>Data analysis manager<br>Regional sales manager<br>Software sales manager<br>Data management manager |
| **Cluster 10** | 317 | product manager, data, big data, advanced, direction, platform, product, finance, data analysis, data center | Data product manager<br>Product manager<br>Big data product manager<br>Senior data product manager<br>Senior product manager |

Table 4. The proportion of job demand in each city.

| City category | Percentage of demand | City name | Proportion of demand per city |
|---|---|---|---|
| **First-tier cities** | 26.43% | Shanghai | 7.03% |
| | | Guangzhou | 6.77% |
| | | Shenzhen | 6.41% |
| | | Beijing | 6.23% |
| **New first-tier cities** | 55.43% | Xi'an | 6.97% |
| | | Nanjing | 6.80% |
| | | Hangzhou | 6.63% |
| | | Wuhan | 6.16% |
| | | Chengdu | 6.08% |
| | | Tianjin | 5.86% |
| | | Zhengzhou | 4.39% |
| | | Changsha | 4.03% |
| | | Chongqing | 3.85% |
| | | Shenyang | 2.24% |
| | | Hefei | 1.92% |
| | | Wuxi | 0.49% |
| **Others** | 18.14% | Jinan | 5.75% |
| | | Dalian | 2.60% |
| | | Changchun | 2.18% |
| | | Guiyang | 1.95% |
| | | Xiamen | 1.79% |
| | | Shijiazhuang | 1.58% |
| | | Fuzhou | 1.05% |
| | | Ningbo | 0.75% |
| | | Harbin | 0.50% |

36.09% of the big data jobs are not limited to experience requirements, but 31.44% of the jobs require 1–3 years of work experience, and 23.92% require 3–5 years of work experience. This indicates that most enterprises have experience requirements for big data work, and at least one year of work experience is required, so experienced job seekers are more popular in the field of big data.

Can be seen from the Fig 2, all kinds of city, the salary range for the 4 k—8 k accounts for large data-position than most, the more developed city jobs pay less than 4 k is less, as the urban economic level has increased the average salary is on the rise, and more than 8 k proportion pay more and more high, bargaining proportion also shows ascendant trend, suggesting that big data post salary is associated with the city's economic development level.

## 5.2 The characteristics of big data jobs from the whole data set

As shown in Fig 3 Category 1 accounts for 14.031% of the job demand of big data, mainly engaged in data development, research and development and other work. Java and database, and other words appear in the clustering keywords, indicating that big data development post requires job seekers to master a development language and database skills. Category 2 is mainly for a data engineer, big data operation and maintenance engineer, etc., with a total of 994 positions and less demand. Data operation is a keyword in category 3, and the demand for big data positions is not high. There are only 756 positions in category 4, which accounts for a small proportion of the demand for big data positions. It is mainly about basic data entry and

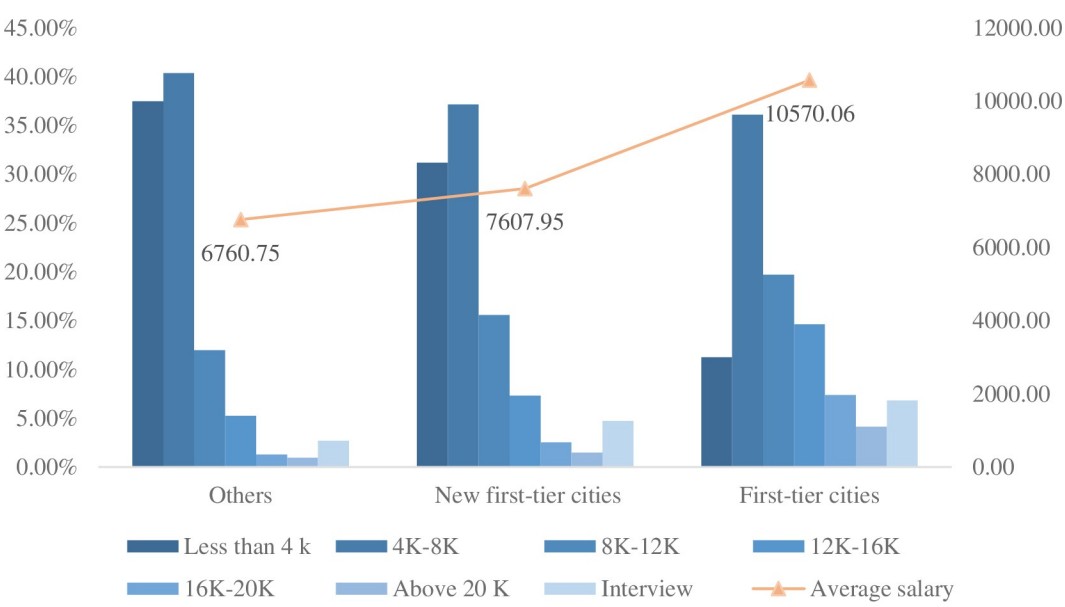

**Fig 2. Salary structure chart of various cities.** K stands for one thousand RMB, the proportion of different salary stages in different city categories.

management, which requires job seekers to master certain database skills. Category 5 has a total of 1,465 positions, accounting for 10.106%, ranking the third in the job demand of big data, mainly engaged in data statistics, processing and analysis, and the relevant big data positions are data specialist, data statistics specialist, etc. Category 6 has the largest demand in the recruitment of enterprises, with a total of 5,122 positions, accounting for 35.334% of all positions, mainly engaged in data processing, statistics, marking, and other work, the mainstream positions are data statistician, data clerk, data marker, project manager, data supervisor and so on. There are many overlapping words in the clustering keywords of category 7 and category 8. From the perspective of related high-frequency positions, the two categories have certain similarities, mainly engaged in data analysis, and the mainstream positions are data analysts,

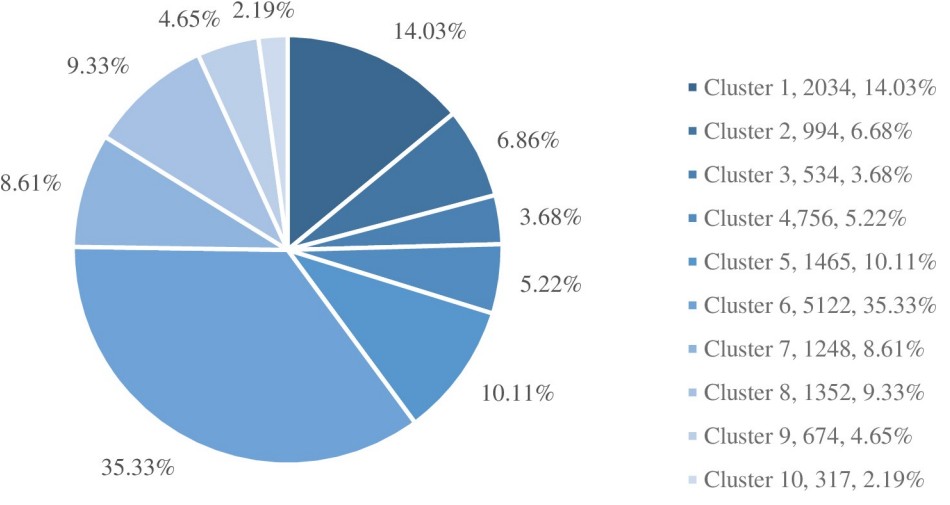

**Fig 3. The proportion of vacancies in all categories.**

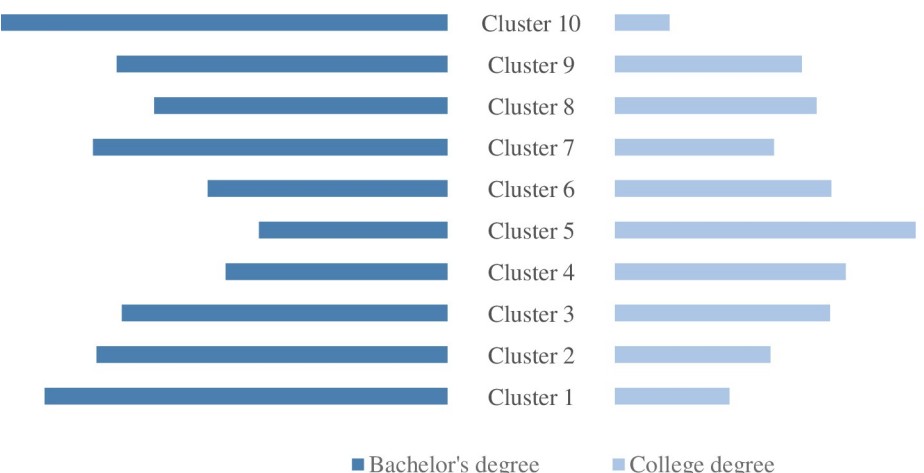

**Fig 4. Comparison of requirements for undergraduate and junior college degrees of various positions.**

data analysis engineers, data analysis specialists, etc., with a total of 2,600 positions, accounting for 17.936% of the total. However, there are some differences between the two categories in the field of data analysis. The high-frequency positions in category 9 and category 10 are all at the manager level. They are high-end talents in the field of big data with higher ranks and fewer positions. They are mainly engaged in data management manager, product manager, sales manager, and other professions.

Different job categories in big data jobs have certain differences in educational background and experience requirements. Combined with the analysis of Figs 4 and 5, the job requirements of category 10 are more inclined to candidates with a bachelor's degree and more than 3 years of experience, because the product manager belongs to the company Executives naturally have higher requirements for experience and academic qualifications. Although category 9 mainly recruits manager-level positions, the sales field has lower requirements for academic qualifications than product managers and has a higher acceptance of college degrees, but they are still more inclined to job seekers with more than 3 years of experience. Category 1 has

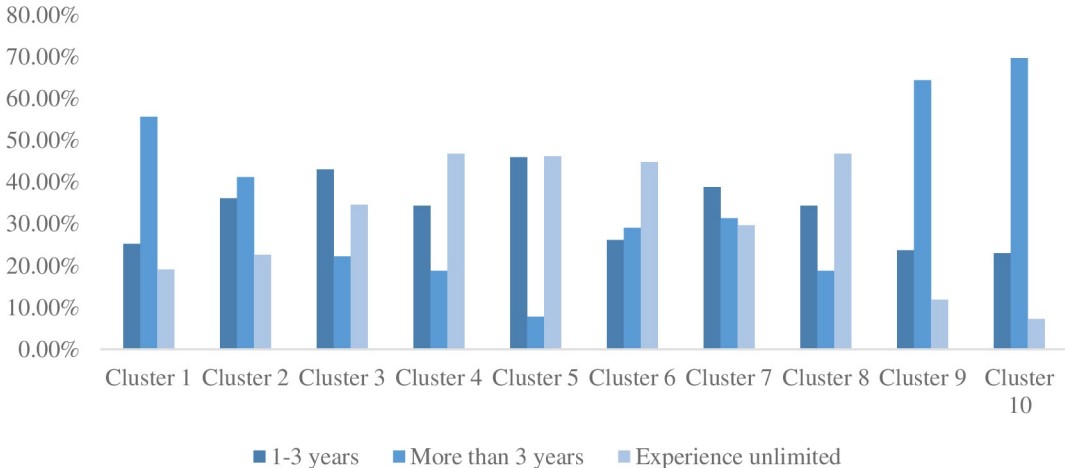

**Fig 5. Different proportions of experience requirements for various positions.** The proportion of three experience requirements in different job categories.

gathered a large number of big data development positions, which have high requirements for experience and academic qualifications. Companies hope to recruit personnel with a bachelor's degree and more than 3 years of development experience. From the previous analysis, the difference in job fields between categories 7 and 8 is still reflected in academic qualifications and experience. Category 7 involving the financial field has higher requirements for academic qualifications and experience than category 8 in the sales field and the threshold for recruitment Relatively high. Category 6 contains a wide range of job categories. The demand for academic qualifications is relatively evenly distributed, and the experience requirements are relatively low. Many companies do not make special requirements for the experience of job seekers. From Figs 4 and 5, we can see that categories 2, 3, 4, and 5 have gradually reduced requirements for a bachelor's degree and more than 3 years of experience, and the demand for a college degree and unlimited experience has become higher, which indicates that the more basic the big data work, the lower the threshold for job hunting, such as data entry, data statistics, data labeling, etc. When the job is more difficult and more complex, the higher the requirements for job seekers will be, such as data operation, data analysis specialist, and other positions.

The demand ratio and average salary of each job category in cities are shown in Fig 6. It can be seen from the figure that the demand for product manager positions represented by category 10 in first-tier cities is higher than that in the other two cities, and the average salary is the highest. The demand for other types of jobs in new first-tier cities is higher than that in first-tier cities. The average salary of category 9 is lower than that of Category 10. Although the high frequency of the two types of positions is manager level, category 9 is mainly concentrated in new first-tier cities, whose economic development level is not as good as that of first-tier cities, so the average salary is naturally low. Categories 7 and 8 have been analyzed in the previous article. There is a certain similarity between the two types of jobs, which is also reflected in their city distribution. The demand for the two types of jobs in each city is very small, but the average salary of category 7 is higher than that of category 8. Combined with the previous analysis, it can be seen that the salary of the financial field in the big data analysis position is higher than that in the sales field. The main demand of category 6 is concentrated in the new first-tier cities, and its average salary is similar to the average salary of the overall big data jobs in the

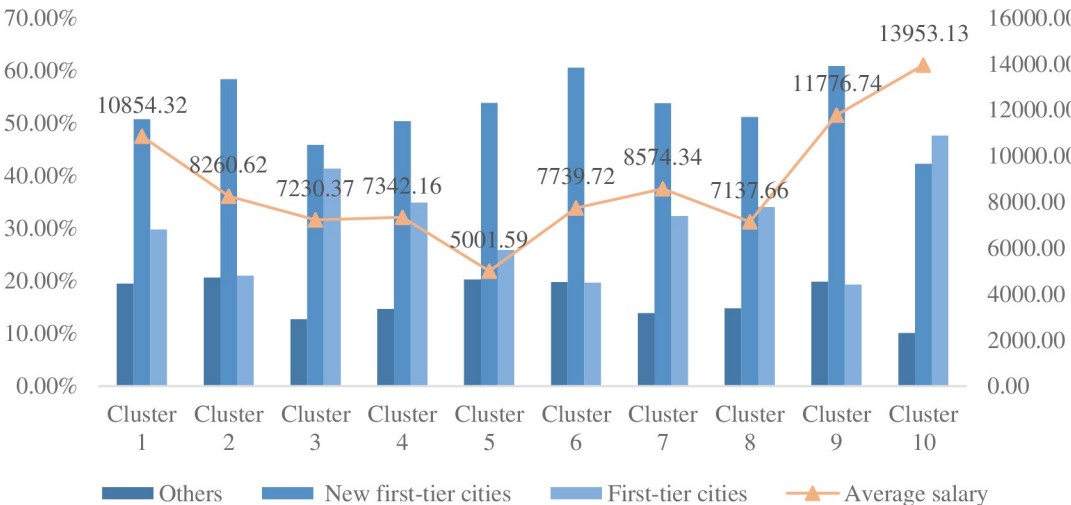

**Fig 6. The proportion of urban demand and the average salary for each job category.** The share of the demand for each job category in each city and the average salary of each job category are calculated based on the RMB income.

new first-tier cities. Category 5 has the lowest average salary. The main reason may be that it is engaged in basic data processing and statistical work. Compared with other types of positions, the job is not difficult and does not require high levels of job applicants. Although the demand for category 4 in first-tier cities is lower than that of category 3, its average salary is higher than that of category 3, indicating that data management and research positions are higher than the salary level of data operations. The demand for category 2 is mainly concentrated in the new first-tier cities, and its salary level is relatively high, but it is lower than that of the big data engineer category 1 of development positions, indicating that the demand for development jobs in big data positions is large and the salary level is high.

## 6. Conclusion

This article uses web crawlers to obtain a large amount of big data job recruitment information, combined with stuttering word segmentation and regular expressions and uses K-means text clustering to divide big data jobs into 10 different categories and explore the needs of big data jobs. After analyzing the data set and clustering results, the following conclusions are drawn. The demand for big-data-related jobs is concentrated in first-tier cities and new first-tier cities. The education threshold is relatively low, with bachelor's degree and junior college degree as the major, and the proportion of doctor's degree and master's degree is very small. On the one hand, this is because the population base of highly educated people is small; on the other hand, many domestic universities have not started the doctoral and master's degree programs related to big data. Secondly, from the perspective of the big data industry as a whole, enterprises are more inclined to recruit job seekers with at least one year's experience. In some basic job categories, the requirements for experience will be reduced, while for high-level management and development positions, there will be higher requirements for experience. The salary level of a big data post is related to its city type and job position. On the one hand, the economic development of a city affects the salary level; on the other hand, the rank and job content of a post determine the salary level of the post. In the clustering results, the demand for manager-level posts is low, the average salary is high, and the requirement for their education and experience is relatively high. Compared with the manager position, the demand for development posts is relatively high, and the salary level is relatively high compared with other categories, but the requirement for experience is high, and there is a certain educational threshold. However, the data processing, analysis, management, research, operation and other posts have relatively low requirements for education and experience, so the salary level is relatively low.

There are few pieces of research on the job demand of big data related to text mining in China. This paper uses text mining to analyze the job demand characteristics of big data from the aspects of the city, salary, education background, and experience, etc., to clarify the job demand status of the big data industry, provide enterprises and job seekers with the information of the big data demand market, and provide support for the citation research of text mining in various fields. However, there are still some deficiencies in this study. Firstly, due to the uneven information release format on recruitment websites, some data will be omitted during the collection process, and many difficulties will be caused in the data pre-processing process, which will have a certain impact on the subsequent clustering effect. Secondly, there are categories with very unclear job definitions in the clustering results. This article does not further divide and dig into these categories, which may omit some other demand characteristics in the big data job requirements. In the future, we will combine more text mining technologies, such as the LDA model, Word2Vec model, etc. to further mine big data job recruitment information, and analyze the technology, skills, and knowledge required under different job categories.

## Supporting information

**S1 Data. Minimal data new.**
(XLSX)

## Author Contributions

**Data curation:** Ma Yinxia.

**Methodology:** Ma Yinxia.

**Resources:** Dai Debao, Zhao Min.

**Software:** Ma Yinxia.

**Supervision:** Dai Debao, Zhao Min.

**Writing – original draft:** Ma Yinxia.

**Writing – review & editing:** Dai Debao.

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
