## [Decision Letter · Decision Letter 0]

13 Apr 2021

PONE-D-21-07023

Analysis of Big Data Job Requirements based on K-means Text Clustering in China

PLOS ONE

Dear Dr. yinxia,

Thank you for submitting your manuscript to PLOS ONE. After careful consideration, we feel that it has merit but does not fully meet PLOS ONE’s publication criteria as it currently stands. Therefore, we invite you to submit a revised version of the manuscript that addresses the points raised during the review process.

We look forward to receiving your revised manuscript.

Kind regards,

Bing Xue, Ph.D.

Academic Editor

PLOS ONE

Journal Requirements:

Reviewers' comments:

Reviewer's Responses to Questions

**Comments to the Author**

1. Is the manuscript technically sound, and do the data support the conclusions?

Reviewer #1: Yes

Reviewer #2: Yes

2. Has the statistical analysis been performed appropriately and rigorously? 

Reviewer #1: Yes

Reviewer #2: Yes

3. Have the authors made all data underlying the findings in their manuscript fully available?

Reviewer #1: No

Reviewer #2: No

4. Is the manuscript presented in an intelligible fashion and written in standard English?

Reviewer #1: Yes

Reviewer #2: Yes

5. Review Comments to the Author

Reviewer #1: Dear Authors.

Thanks for the opportunity given me to read your Manuscript with Title: "Analysis of Big Data Job Requirements based on K-means Text Clustering in China" I found the paper very interesting.

In your Methodology where data was collected from Zhaopin.com recuitment website by using web crawler software to capture the recruitment information of big data. However you further stated the preprocessed data and 14,496 valid data were obtained after deleting invalid and incomplete data.

My question is can you make all the data available and clear. Also the process how the data was preprocessed.

Thank you for the good job.

Reviewer #2: The introduction section lacks in problem statement and motivation of the study. There is immense need to give arguments in favor of conducting this study. In the conceptual model and hypothesis, it is highly recommended to support the arguments with the support of theory.

6. PLOS authors have the option to publish the peer review history of their article (what does this mean?). If published, this will include your full peer review and any attached files.

Reviewer #1: No

Reviewer #2: No

---

## [Author Response · Author response to Decision Letter 0]

24 May 2021

Responds to the reviewer’s comments:

Reviewer #1:

1. Response to comment: Make all the data available and clear

Response: Considering the Reviewer’s suggestion, we have added Table 1 on the basis of the original text to show the data we collected, and listed specific examples in 3.1 data collection, so that readers can better understand the content of the data we collected. 

2. Response to comment: How the data was preprocessed

Response: We are sorry that we did not show the detailed data preprocessing process. In the revised version, we added Figure 1 to illustrate the specific data processing process in 3.2 data procession.

Special thanks to you for your good comments.

Reviewer #2:

1. Response to comment: The introduction section lacks in problem statement and motivation of the study

Response: We have re-written this part according to the Reviewer’s suggestion. We restate the question and motivation of the study in the introduction.

2. Response to comment: There is immense need to give arguments in favor of conducting this study

Response: It is really true as Reviewer suggested that this article lacks theoretical basis. Therefore, in the new version, we add three references as the theoretical basis and support of the research.

Special thanks to you for your good comments.

We tried our best to improve the manuscript and made some changes in the manuscript. These changes will not influence the content and framework of the paper. And here we did not list the changes but marked in red in revised paper.

We appreciate for Editors/Reviewers’ warm work earnestly, and hope that the correction will meet with approval.

Once again, thank you very much for your comments and suggestions.

---

## [Decision Letter · Decision Letter 1]

16 Jul 2021

Analysis of Big Data Job Requirements based on K-means Text Clustering in China

PONE-D-21-07023R1

Dear Dr. yinxia,

We’re pleased to inform you that your manuscript has been judged scientifically suitable for publication and will be formally accepted for publication once it meets all outstanding technical requirements.

Kind regards,

Bing Xue, Ph.D.

Academic Editor

PLOS ONE

Additional Editor Comments (optional):

Reviewers' comments:

Reviewer's Responses to Questions

**Comments to the Author**

1. If the authors have adequately addressed your comments raised in a previous round of review and you feel that this manuscript is now acceptable for publication, you may indicate that here to bypass the “Comments to the Author” section, enter your conflict of interest statement in the “Confidential to Editor” section, and submit your "Accept" recommendation.

Reviewer #1: All comments have been addressed

Reviewer #2: (No Response)

2. Is the manuscript technically sound, and do the data support the conclusions?

Reviewer #1: Yes

Reviewer #2: Yes

3. Has the statistical analysis been performed appropriately and rigorously? 

Reviewer #1: Yes

Reviewer #2: I Don't Know

4. Have the authors made all data underlying the findings in their manuscript fully available?

Reviewer #1: Yes

Reviewer #2: Yes

5. Is the manuscript presented in an intelligible fashion and written in standard English?

Reviewer #1: Yes

Reviewer #2: Yes

6. Review Comments to the Author

Reviewer #1: Dear Authors,

Thank you for your great work. I reccommend this paper with Title: Analysis of Big Data Job Requirements based on K-means Text Clustering in China for publication.

Reviewer #2: The manuscript appears to be an interesting study and will be a valuable contribution to the relevant field of study. I wish the authors all the best.

7. PLOS authors have the option to publish the peer review history of their article (what does this mean?). If published, this will include your full peer review and any attached files.

Reviewer #1: No

Reviewer #2: No

---

## [Editor Report · Acceptance letter]

27 Jul 2021

PONE-D-21-07023R1 

Analysis of Big Data Job Requirements based on K-means Text Clustering in China 

Dear Dr. Yinxia:

I'm pleased to inform you that your manuscript has been deemed suitable for publication in PLOS ONE. Congratulations! Your manuscript is now with our production department. 

Kind regards, 

on behalf of

Professor Bing Xue 

Academic Editor

PLOS ONE